# Changes in Students’ Perceptions Regarding Adolescent Vaccinations through a Before–After Study Conducted during the COVID-19 Pandemic: GIRASOLE Project Study

**DOI:** 10.3390/vaccines11101524

**Published:** 2023-09-25

**Authors:** Vincenzo Restivo, Alessandra Bruno, Giuseppa Minutolo, Alessia Pieri, Luca Riggio, Maurizio Zarcone, Stefania Candiloro, Rosalia Caldarella, Palmira Immordino, Emanuele Amodio, Alessandra Casuccio

**Affiliations:** 1Department of Health Promotion, Maternal and Infant Care, Internal Medicine and Medical Specialties (PROMISE) “G. D’Alessandro”, University of Palermo, Via del Vespro 133, 90127 Palermo, Italy; alessandra.bruno05@unipa.it (A.B.); giuseppa.minutolo@unipa.it (G.M.); luca.riggio@unipa.it (L.R.); stefania.candiloro@unipa.it (S.C.); palmira.immordino@unipa.it (P.I.); emanuele.amodio@unipa.it (E.A.); alessandra.casuccio@unipa.it (A.C.); 2Clinical Epidemiology and Cancer Registry Unit, University Hospital “P. Giaccone”, Via del Vespro 133, 90127 Palermo, Italy; alessia.pieri@policlinico.pa.it (A.P.); maurizio.zarcone@policlinico.pa.it (M.Z.); rosalia.caldarella@policlinico.pa.it (R.C.)

**Keywords:** protection motivation theory, vaccination knowledge, vaccine, adolescents, intervention, socio-economic level, online, before–after, COVID-19, communication

## Abstract

The COVID-19 pandemic caused a reduction in vaccination coverage for all age groups, especially in non-infant age. The main objective of the present study is to evaluate the effectiveness of an online intervention conducted among adolescents during the COVID-19 pandemic in increasing knowledge and positive attitudes toward vaccinations. The study, which took place online from March to May 2021, involved 267 students from six lower secondary schools in Palermo city (Italy); they filled out the questionnaire before and after the intervention. The questionnaire was based on the protection motivation theory (PMT), which estimates the improvement in vaccination-related knowledge and attitudes. The pre- and post-intervention comparison showed a significant increase in the perception of the disease severity: strongly agree pre-intervention *n* = 150 (58.6%) and post-intervention *n* = 173 (67.6%, *p* < 0.001), rated on a five-point Likert scale. In a multivariate analysis, the factor associated with the improvement in the score after the intervention was the school dropout index (low vs. very high dropout index OR 4.5; *p* < 0.03). The educational intervention was more effective in schools with lower early school leaving rates, an indirect index of socio-economic status. The topic of vaccination has caught the adolescents’ attention, it is, therefore, important that interventions tackling teenagers are tailored to reduce their emotional tension about the perception of adverse effects and improve vaccination coverage.

## 1. Introduction

Worldwide, the pandemic caused by severe acute respiratory syndrome coronavirus 2 (SARS-CoV-2) and stay-at-home orders have caused an abrupt drop in adherence for routine immunization in all milestone age cohorts, especially in adolescents, because both demand and supply have slowed down. The Italian Society of Pediatrics carried out a survey among 1500 parents, revealing that over 33% of them postponed childhood scheduled vaccinations due to fear of COVID-19 [1]. Likewise, in Italy, vaccination coverage significantly declined from new-borns to elderly people. In particular, in 2021, Italian data showed a significant worsening in vaccination coverage for tetanus, diphtheria, pertussis, and polio (Tdap-IPV) in the 2003 and 2005 cohorts (respectively, 18 and 16 year olds), undermining the improving trend that occurred after 2017. As an example, the vaccination coverage for the fifth dose of Tdap-IPV decreased from 74.28% in 2019 to 73.23% in 2021 for the 18-year-old cohort [2]. On the one hand, people avoided healthcare settings for fear of contagion and to comply with the stay-at-home orders; on the other hand, vaccination services, complying with containment measures, gave priority to anti-SARS-CoV2 vaccines and new-born vaccinations, resulting in decreased accessibility to the administration of teenager-targeted vaccines [1]. Therefore, COVID-19 negatively affected routine immunization, potentially determining a reoccurrence of vaccine-preventable diseases (VPDs) when social contact increased again.

Since teenagers were among the most affected, there is a need to reach higher vaccination coverage. A major role should be played by primary care providers, such as pediatricians and general practitioners (GPs), mainly in areas of low uptake and among vulnerable groups, who are more susceptible to misinformation and could benefit more from counseling with trusted healthcare workers [3,4].

School-based educational interventions for adolescents are considered the most effective means to ensure high vaccine coverage for adolescents, since they improve attitudes, intentions, and uptake towards vaccinations [5]. In Australia, school-based educational programs have achieved higher vaccination coverage, and schools have become the primary setting for the delivery of vaccinations to adolescents, especially given the decline in the frequency of visits to primary care practitioners of adolescents compared with children [5]. In Italy, the school medicine service was founded in the 1960s and played an important role in preventing diseases and promoting health and vaccination programs for school students. However, the program lost its functions and applications and slowly disappeared following the reforms of the 1990s, which transformed the Local Health Units (USLs) into public enterprises (ASLs) [6].

School closures during the first wave of the COVID-19 pandemic exerted considerable effects on immunization in countries where routine immunizations used to be delivered in school settings [7]. Furthermore, face-to-face interventions could not be conducted during the COVID-19 pandemic and new strategies had to be implemented. During lockdowns, online surveys have been shown to be a suitable and useful method for health promotion interventions, especially when face-to-face research is restricted. Self-reported online tools have indeed proven to provide realistic assessments of vaccine literacy levels [8].

The primary objective of this study was to evaluate, through the administration of a before–after questionnaire, the efficacy of an online educational intervention regarding knowledge and attitudes towards vaccinations in adolescents during COVID-19 restrictions.

## 2. Materials and Methods

### 2.1. Study Design and Population Involved

This before–after study consisted of an educational intervention, carried out from March to May 2021 through distance-learning synchronous teaching platforms, on a sample of adolescents from lower secondary schools in the city of Palermo, Italy. Parents were also invited to participate with or without their children. The project was named “Giornata Informativa sulle RAccomandazioni per le vaccinazioni in Sicilia dedicata agli adOLEscenti (G.I.RA.S.OLE)”. Overall, 23 lower secondary schools of Palermo city, representing different socio-economic backgrounds, were approached by contacting the dean of each school, and 6 of these (26%) accepted to participate in the study. The study population included 8554 students attending one of the 23 lower secondary schools of Palermo, Italy.

The inclusion criteria were that individuals had to (i) be attending a secondary level school in the city of Palermo during the survey period, (ii) be aged between 10 and 17 years, (iii) parents or legal tutors had to agree to the students’ participation as those individuals were minors, and (iv) they had to fill in the questionnaire before and after the intervention.

The Ethics Committee of Palermo 1 approved the study on 20 January 2021.

### 2.2. Intervention

The intervention consisted of an online information campaign conducted through an online platform by healthcare workers skilled in adolescents’ vaccinations. It focused on epidemiological information on VPDs, vaccine accessibility, as well as the safety and effectiveness of vaccination. It was based on adolescent-targeted vaccinations recommended by the Sicilian vaccination schedule, such as Tdap-IPV, HPV, meningococcal ACWY, and B vaccines.

In detail, the intervention lasted about 30 min, during which the following information was provided:-counselling about adolescent-targeted vaccines;-the most common VPDs among teenagers;-benefit–risk profile of vaccines.

Afterward, an online question and answer section was made available to have an interactive group discussion and discuss any doubts about vaccinations and VPDs. This section was organized through an online anonymous chat using the tools of the online platform.

The effectiveness of the intervention was assessed through the administration of a questionnaire before the intervention (pre-intervention questionnaire) and after its conclusion (post-intervention questionnaire).

### 2.3. The Questionnaire

The research tool was a questionnaire, created on an online platform, consisting of three sections evaluating the socio-demographic characteristics of students and their parents or legal tutors, their vaccination history, and the vaccination motivation of children using the protection motivation theory (PMT) as a theoretical framework.

The first part of the questionnaire addressed socio-demographic factors that could influence the individual’s behavior towards vaccination, such as gender (male or female), age (years), school institution and class attended, place of residence (rural or urban), and the dropout school index.

Southern Italy suffers from one of the highest rates of early school leavers in Europe, as shown by the last EUROSTAT report [9]. Accordingly, each of the schools involved in the study has been linked to the dropout school rate of the specific neighborhood, using data from the Sicilian Department of Education. This index evaluates the differential rate between the students enrolled in the 1st year and those in the 3rd year of lower secondary schools in the neighborhood [10]. The participating schools were classified into four groups according to the dropout school index of the neighborhood in which the school was located: low, middle, high, very high.

The second part of the questionnaire included questions on the immunization history of students: the last vaccination received, and the age at which it was received.

The third part of the questionnaire was written following the protection motivation theory, a model developed by Rogers in 1975 which describes how individuals are motivated to behave in a self-protective way against a perceived health threat (e.g., VPDs) [11]. This framework has previously been used in other studies to evaluate willingness to receive vaccines, such as measles, mumps, and rubella [12], HPV [13], and influenza [14,15]. Nevertheless, these previous studies used the PMT model only on adults. However, there are other studies, involving adolescents aged 11 to 17 years, which have been conducted with the aim of assessing their knowledge, attitude, and perception of vaccination pre- and post-educational interventions. Nonetheless, these last studies on adolescents did not use PMT: they focused mainly on HPV vaccination and used different theories, such as the health belief model (HBM), the theory of reasoned action (TRA), the theory of planned behavior, and social cognitive theory (SCT) [5].

PMT is a theoretical framework which suggests that the two main cognitive processes which influence people’s motivation to protect themselves when faced a threatening event are threat appraisal and coping appraisal. Threat appraisal and coping appraisal are two parallel cognitive processes which determine intention to engage in a recommended health behavior, and intention is the most proximal predictor of behavior [11]. PMT cognition domains are perceived severity of the negative consequences of health threats; perceived susceptibility to the negative consequences of health threats; response efficacy, which refers to one’s realization that the preventive behavior is effective against the negative consequences of the health threats; self-efficacy, defined as the belief that the individual can appropriately perform the preventive behavior; maladaptive response rewards, that is, the benefits in case the individual does not perform the preventive behavior; perceived response cost, represented by the barriers to perform the preventive behavior (effort, time, money); and intention to engage in the referent behavior [11]. Coping and threat appraisals are, respectively, associated with adaptive and maladaptive responses. PMT emphasizes the role of cognitive mediating processes in health behavior change. The proposed PMT model is shown in Figure 1.

Beliefs and attitudes towards vaccinations were assessed according to the answers to seven questions (“How much do you agree with the following statement?”), one for each PMT construct (Table 1). All constructs of PMT were rated on a five-point Likert scale: 1 = “strongly disagree”, 2 = “tend to disagree”, 3 = “hard to say/undecided”, 4 = “tend to agree” and 5 = “strongly agree”. Based on the meaning of the phrases, some questions were scored in the opposite way, to have all answers as increasing scores.

In order to comply with COVID-19 pandemic restrictions, the questionnaires were self-administered by the students of the individual classes through a digital online platform. Some school regulations did not allow students to use tablets or mobile phones, and for this reason they filled out conventional paper questionnaires. Both the online and the paper versions of the questionnaire were anonymous.

### 2.4. Statistical Analysis

Absolute and relative frequencies were reported for qualitative variables. The skewness and kurtosis test assessed the distribution of quantitative variables: mean and standard deviation (SD) for normally distributed variables, and median and interquartile range (IQR) for those non-normally distributed. The comparison of medians between pre- and post-interventions was performed by means of the Wilcoxon test.

A univariable logistic regression analysis was performed to evaluate the factors associated with an increase in the PMT score before and after the intervention, that was a summary measure of the before after questionnaires for each subject. The variables included in the univariable analysis were the student’s age, the parent’s age, sex, school grade, location of the household, school dropout rate, and last vaccination received.

The multivariable logistic regression model of factors associated with PMT increase before and after the intervention used a backward stepwise selection to include other associated variables according to the likelihood ratio test for different models. Furthermore, age, sex, and course year were included as a priori potential confounders. Crude and adjusted odds ratios (cOR and aOR, respectively) and the related 95% confidence intervals (95% CI) were reported in the final model, as well as the *p*-value. For all analyses, a *p*-value below 0.05 was assumed to indicate statistical significance (two tailed).

A multivariate imputation by chained equations (MICE) was applied to deal with the missing data in the sample [16]. The method is based on fully conditional specification, where each incomplete variable is imputed by a separate model. Imputation of missing data was carried out with the package “mice” (version 3.7.0) using RStudio and the statistical software R (version 3.6.2), following the proportional odds model option [17]. Structural equation modeling with the estimator of diagonally weighted least squares (robust variant of weighted least squares using the Satorra–Bentler scaled test statistic method to correct the model and to produce robust standard errors) was set to test if the collected data fit with the proposed model, using the package “lavaan” (0.6) for R (version 3.6.2). All other analyses were performed using the STATA v14.2 software.

## 3. Results

Overall, 256 out of 427 participating students answered the survey before and after the intervention, with a participation rate of 60.0%. The participants’ characteristics are shown in Table 2. The majority of the respondents were males (53.5%, *n* = 137) and 92.2% (*n* = 236) were living in the metropolitan area of Palermo. The median age of all students was 13 years (IQR = 12–14). According to the school dropout index of the neighborhood, 31% (*n* = 80) of the responders attended the courses with low school dropout index, 21% (*n* = 54), medium, 15% (*n* = 37) high, and 33% (*n* = 85) very high.

Approximately half of the surveyed students (56.5%, *n* = 140) could not remember the last vaccination they received. Among the subgroup who remembered, only 11.5% (*n* = 16) declared that they had received it at less than 5 years old, and only 5.0% (*n* = 7) asked their parents to be brought to the vaccination service. The participants reported that the last received vaccines were meningococcus (29.6%, *n* = 21), HPV (12.7%, *n* = 9), and influenza (12.7%, *n* = 9).

The results of the path analysis (Figure 2) showed that coping appraisal is the most powerful predictor of the protecting motivation, with β = 0.750 on PMT score increase, followed by coping appraisal β = 0.598 on intention and threat appraisal, β = 0.536 on PMT score increase.

As reported in Table 3, the comparison of all the PMT items between pre- and post-intervention showed an increase in the individual PMT score for 27% (*n* = 69) of students after the intervention. In detail, the item with the highest increase was the response efficacy, which increased for 23.1% of students (*n* = 61). This was followed by intention, with an increase of 21.1% (*n* = 54), and perceived severity, with an increase of 18.4% (*n* = 47) (Table 3).

Logistic regression analysis (Table 4) assessed the association with socio-demographic factors among people who increased their PMT score after the intervention.

The school dropout rate of the neighborhood in which the school is located was the only social factor that significantly influenced the improvement of the PMT scores after the intervention by controlling for sex, age, school grade, age of mothers, history of previous vaccination, and location of the household. The school dropout rate also had a decreasing trend from high (OR = 8.3, *p* = 0.009) to low (OR= 4.5 *p* = 0.03).

## 4. Discussion

This study evaluated the effectiveness of an online informative education intervention about vaccination carried out among adolescents. There has been a wide body of research in the past years studying the efficacy of pre- and post-educational classes about vaccination in adolescents, focusing mainly on HPV, Tdap-IPV, MMR, and influenza [18]. However, to our knowledge, there has not been a pre–post-education intervention that focuses on adolescent-targeted vaccinations and uses PMT as a framework.

Several studies examined all the factors associated with the intention to receive vaccines according to PMT. These findings show that the intention to vaccinate is positively correlated with four parameters (severity, susceptibility, response efficacy, and self-efficacy) and negatively with two parameters (maladaptive response rewards and perceived response costs) [12,13]. A systematic review confirmed that perceived severity, vulnerability, response efficacy, self-efficacy, and low maladaptive response rewards contributed to increasing the motivation to receive the influenza vaccine during the 2009 pandemic [15]. This manuscript highlights that greater intentions were significantly associated with higher coping appraisal (Coef.  =  0.598, *p* < 0.001) and threat appraisal (Coef.  =  0.312, *p* = 0.03). The significant correlation between coping appraisal and vaccination intention is in line with the international literature, which suggests that coping appraisal typically has a greater influence on motivation intention than threat appraisal [11]. This research provides further evidence of the importance of self-efficacy, perceived response cost (*p* < 0.001), and response efficacy (*p* = 0.003) in predicting intention to vaccinate, consistent with previous research that has applied these coping appraisal constructs to the prediction of vaccination uptake [15]. Self-efficacy is considered a great predictor of health-related behavior change and maintenance, and individuals with self-efficacy to vaccination tend to develop vaccination intention [19,20]. Similarly, in this study, self-efficacy is one of the main drivers of vaccine acceptability and of a lower role for severity of disease perception. Furthermore, response efficacy is associated with vaccine availability, affordability, and accessibility. However, the main barrier in Italy is not economic, since teenagers’ vaccinations are offered free of charge, but may be related to the discontinued provision of immunization services during the COVID-19 pandemic [1]. However, the key barriers to teenagers receiving vaccinations appears to be concerns about side effects and safety. Moreover, a fundamental role in determining vaccination intention is played by threat appraisals (Coef. = 0.312, *p* = 0.003), defined as the strategies used to evaluate the risk of a health threat. Addressing individuals’ perceived severity and perceived susceptibility may be effective ways to provide motivation for them having a vaccination. Conversely, exposure to news of fear may intensify anxiety and provoke denial, especially in a vulnerable population like adolescents, worsening the mistrust towards authorities.

In the present study, the only socio-demographic factor associated with an increase in the PMT score was the attendance at a school with a higher early dropout rate. School dropout, defined as a major public health challenge, is a concern in Western countries and is associated with lower socio-economic and cultural conditions compared to the native background [21]. The role of the education level as a determinant of vaccine hesitancy has been extensively examined in the literature, and higher socio-economic levels have been associated with more confidence in vaccine safety [22,23]. In addition, a wide range of research has explored how family income and school achievements, as well as social and structural features of neighborhoods, may play significant roles in adolescent health literacy [5]. Findings from a British study about the willingness of adolescents to receive a COVID-19 vaccination showed that vaccine hesitancy was higher among students from a deprived socio-economic context and whose school locations were in areas of greater deprivation [24]. An Australian study about HPV school-based vaccination has provided indicative evidence that lower school attendance and higher socio-economic disadvantage are strongly associated with lower HPV coverage [5]. Such results can be explained in two different ways. Firstly, people with lower educational levels have more difficulty understanding and evaluating information about risks and benefits of vaccines, so they tend to misinterpret them. Furthermore, the COVID-19 pandemic has caused the spread of false and misleading information in digital and physical environments (infodemic), thus causing a negative impact on the behavior toward all recommended vaccinations [25]. Families belonging to a disadvantaged socio-economic neighborhood may lack the tools to facilitate critical thinking as well as evaluation skills required to avoid harmful misinformation on the safety and effectiveness of vaccinations, since health literacy is firmly connected with socio-cultural determinants [26]. Misinformation about COVID-19 has caused psychological distress and has resulted in declining trust in the government and in the healthcare system, contributing to vaccine hesitancy in the period in which the data of the present study were collected [20]. The improvement in confidence towards vaccines requires less complex communication by healthcare workers, also considering that social media will play an increasingly important role in promoting communication about health [27]. Secondly, the lower vaccination uptake in disadvantaged socio-economic groups can be imputed to the difficulty of access to vaccination services because of nonfinancial barriers (i.e., availability, accessibility, accommodation, and acceptability, according to the framework theorized by Penchansky and Thomas) [28,29]. Indeed, as stated above, affordability (financial barrier) can be excluded. Disparities and inequities in access have increased during the COVID-19 pandemic because of the prioritization of healthcare services directed towards SARS-CoV-2 care [1,3]. People living in resource-limited settings had indeed to face physical barriers because of transport interruptions, economic hardships, and fewer opportunities to be visited by a GP or pediatrician. This may explain why vaccine hesitancy is higher in deprived communities. To overcome these barriers and increase vaccination uptake, public health decisions should be addressed by tailoring vaccination campaigns using non-complex language for parents and children, having in mind particular groups of the population. As a matter of fact, individual perceived knowledge about a vaccination is a critical factor which enhances adherence to self-protective measures like vaccine uptake, and is significantly correlated with coping and threat appraisals. The correlation between perceived knowledge and vaccination behavior has been corroborated by a study performed during the COVID-19 pandemic exploring factors associated with COVID-19 vaccination [20]. On the one hand, correct knowledge is an effective factor in facilitating the adoption of protection measures in pandemics, on the other hand, misinformation and lack of knowledge regarding COVID-19 vaccines interferes with vaccine uptake and contributes to vaccine hesitancy.

The increased perception of response costs of PMT is a finding that supports the already well-explored evidence that vaccines can potentially generate anxiety, especially among teenagers [30]. There is no doubt that the topic of vaccination can cause discomfort and potential fear, especially in such a period like spring 2021, when a few adverse reactions to COVID-19 vaccines, like atypical thromboses, were first reported and were a key focus of interest by mass media [31,32]. Consequently, the perception of vaccination risks could be higher than the potential benefits of infectious disease prevention, which seem to be just slightly perceived. Furthermore, previous studies have demonstrated that adolescents’ concerns about vaccinations are limited to immediate and not long-term adverse effects, such as pain at the injection site, focusing mainly on the needle [33,34]. There is evidence about the magnitude of the impact of needle fear on non-compliance with vaccinations, and literature has reported a significant relationship between the level of needle fear and vaccine non-compliance for children [35]. Likewise, a review has concluded that 4–26% of healthcare workers refused influenza vaccination because of fear of injections [36]. A possible solution to decrease needle fear is to reduce vaccination-related anxiety and to promote adolescent confidence in vaccination demonstrating that the benefits outweigh the fear of adverse events.

Moreover, the survey answers about vaccination history showed that only a few participants paid attention to the vaccinations they had received, while most of them lacked awareness about recommended vaccines. It can be assumed that parents did not involve them in the decision-making process about vaccinations and healthcare decisions in general. This finding is in line with those observed in previous studies, which explored the role of adolescents in vaccine decision making to improve the immunization rate [28]. Adolescents are indeed in a dynamic transition phase between childhood and adulthood, in which effective programs of health promotion can lead to the development of health behavior patterns which can improve vaccination compliance and reduce the fear of vaccination-related adverse events [37]. In addition, several studies have shown the desire of adolescents to be better informed to gain a certain degree of autonomy and involvement regarding vaccination uptake, making decisions with their parents [38].

In the light of the above, policymakers should face the disparities and inequities in healthcare access experienced by economically poorer demographic sections and implement catch-up activities [39,40]. Efforts need to be made to promote integration between the public education system and the national health service by incorporating science classes on health education in school curricula. Suitably tailored interventions using comprehensive language should be the correct method to discuss concerns about vaccine safety and efficacy [41]. Similarly, the administration of vaccines in school settings should be considered, since this method is associated with higher completion rates and raises opportunities to vaccinate adolescents, a group with lower GP or pediatrician attendance than other age groups [42].

The COVID-19 pandemic has caused a sudden and profound disruption in school learning, raising the question of whether internet-based learning is as beneficial as traditional learning. Online interventions, although not always suitable for everyone, allowed new skills to be learned even during lockdowns and other degrees of social restrictions. The internet-based intervention of the present study has engaged a large proportion of students in a short period of time, providing a certain flexibility and allowing a high number of interventions to be performed in schools located in different neighborhoods of the city. As reported in the literature, the benefits of internet-based and face-to-face interventions are comparable [43]. Further studies are necessary to understand the nature of vaccine hesitancy among teenagers in Italy, and the need to implement educational initiatives in order to influence attitudes at an earlier stage of life.

### Limitations

As for any survey-based study, the present work has several limitations. Firstly, the missing data reported in the questionnaire can be a possible source of bias. In order to obtain more consistent results, we used a MICE model. However, for this method there are few solutions to evaluate the validity of analyses [44]. One of the main approaches is to use a cross-validation method. In detail, we first evaluated the predictive value for the PMT items using the imputed database. This analysis showed a low predictive value for the following variables: “perceived susceptibility” among the before intervention questionnaires and “intention” among the after intervention questionnaires. Subsequently, we performed a goodness of fit test for the ordinal response model, showing a good fitting for these variables. Finally, we can assume that the MICE analysis provided appropriate measures of precision for these missing data, incorporating relevant auxiliary variables. Second, the online educational intervention lacks face-to-face stimulation and provides little opportunity for students to engage in discussions. Third, the investigation was conducted through a self-reporting online questionnaire, which may result in biases (social desirability bias, recall bias) and misrepresentation. Fourth, our sample comes from an urban area of southern Italy, therefore, background knowledge and practices of responders cannot be generalized to adolescents from the rest of the country. Fifth, the study adopted a cross-sectional study design, restricting the evidence regarding causal relationships between the studied variables. Nevertheless, this study showed the potential effect in increasing vaccine acceptance by an intervention conducted online and in a country with a low level of vaccination coverage.

## 5. Conclusions

The findings provided by this study highlight teenagers’ lack of knowledge about VPDs and poor involvement in vaccination decision making, and suggest that vaccination knowledge improvement in an online setting among teenagers was strictly associated with the school dropout rate of their neighborhood. School educational programs should focus on enhancing the teenagers’ knowledge and attitudes about VPDs, but at the same time they should address the vaccine-related fear and anxiety deriving from short- and long-term adverse effect of vaccinations. For this reason, stronger communication efforts and school education opportunities are required to promote vaccine decision making among adolescents. Indeed, this study was performed in a period of mistrust and lack of confidence in vaccines, mostly related to the contradictory information spread by the media.

Furthermore, improving healthcare workers’ communication should be one of the most highly prioritized goals when it comes to ensuring timely and complete vaccination of young people, especially for those adolescents who have a vulnerable socio-economic status. Providers, especially in primary care settings, should be trained in effective vaccine communication as well as receive education regarding vaccine recommendations, contraindications, and common concerns of parents and teenagers. The evidence derived from this study highlights the importance of building tailored educational interventions to reduce disparities in vaccination intention among adolescents.

## Figures and Tables

**Figure 1 vaccines-11-01524-f001:**
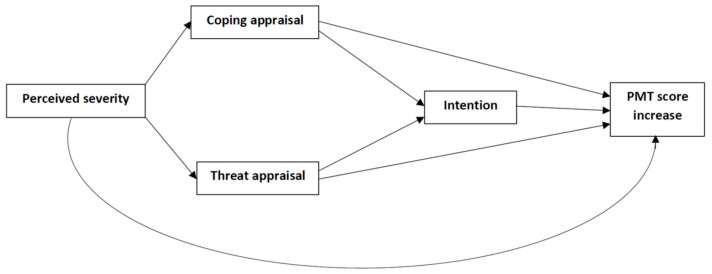
Proposed model illustrated using protection motivation theory to explain vaccine attitudes (PMT score increase).

**Figure 2 vaccines-11-01524-f002:**
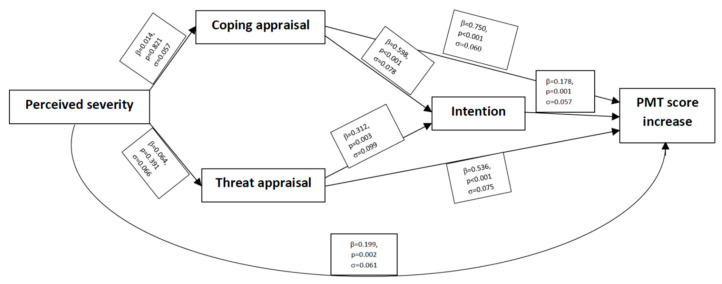
Path analysis of PMT factors.

**Table 1 vaccines-11-01524-t001:** The PMT model applied to the context of adolescent-targeted vaccinations.

Perceived severity	Vaccine-preventable diseases (such as measles, mumps, rubella, chicken pox, polio, tetanus, diphtheria, pertussis, meningitis, and HPV-related diseases) can have severe consequences on my health and can sometimes be fatal. *Example: poliomyelitis causes paralyses to respiratory muscles and death without the artificial lung supplement, Papillomavirus causes cancer to the genital apparatus, measles causes subacute sclerosant panencephalitis, which leads inevitably to death.*
Perceived susceptibility	If I’m not vaccinated, I could suffer from these vaccine-preventable diseases (measles, mumps, rubella, chicken pox, polio, tetanus, diphtheria, pertussis, meningitis, and diseases caused by Papillomavirus). *Example: people who do not get vaccination are not protected against infectious diseases and may have the diseases more frequently than vaccinated subjects.*
Maladaptive response rewards	Vaccination against these vaccine-preventable diseases (measles, mumps, rubella, chicken pox, polio, tetanus, diphtheria, pertussis, meningitis, and HPV-related diseases) causes me to skip school days (because the vaccination centre is far and/or has restricted working hours) and/or can make me sick. *Example: to get to the vaccination centre, I must be absent from school and/or reduce time spent with my friends.*
Self-efficacy	Vaccination protects me from these vaccine-preventable diseases (measles, mumps, rubella, chicken pox, polio, tetanus, diphtheria, pertussis, meningitis, and HPV-related diseases) *Example: infectious diseases have much more severe consequences than adverse reactions to vaccines.*
Response efficacy	Vaccination for these vaccine-preventable diseases (measles, mumps, rubella, chicken pox, polio, tetanus, diphtheria, pertussis, meningitis, and HPV-related diseases) is available for free and there is not any problem for my parents to drive me to the vaccination centre. *Example: many of these vaccines are available for free.*
Perceived response cost	Vaccines for these vaccine-preventable diseases (measles, mumps, rubella, chickenpox, poliomyelitis, tetanus, diphtheria, pertussis, and diseases due to HPV, Meningococcus ACW_135_Y and B) are painful and/or I fear needles. *Example: being vaccines administrated intramuscularly, the fear of needles or pain is upper than the fear of infectious diseases.*
Intention	I want to get a vaccination against vaccine-preventable diseases (measles, mumps, rubella, chickenpox, poliomyelitis, tetanus, diphtheria, pertussis, and diseases due to HPV, Meningococcus ACW_135_Y and B). *Example: you want to get vaccination even if your parents are against it.*

**Table 2 vaccines-11-01524-t002:** Characteristics of the participating students and their vaccination histories.

Adolescent Characteristics	*n* = 256
Age	
<13 years	124 (51.4%)
>13 years	132 (48.6%)
**Sex**	
Male	137 (53.5%)
Female	119 (46.5%)
School grade of secondary school	
1st year	121 (47.3%)
2nd year	31 (12.1%)
3rd year	104 (40.6%)
Location of the household	
Urban area	236 (92.2%)
Rural area	20 (7.8%)
School dropout index	
Very high	85 (33.2%)
High	37 (14.4%)
Medium	54 (21.1%)
Low	80 (31.3%)
Age of mother ^1^	
≥44 years	115 (44.9%)
<44 years	113 (44.1%)
**Can you remember the last vaccination you received? ^2^**	
Yes	108 (42.1%)
No	140 (54.6%)
How old were you when you last received a vaccination? ^3^	
<5 years	16 (11.5%)
5–10 years	63 (45.3%)
>15 years	60 (43.2%)
Did you ask your parents to bring you to the vaccination center?	
Yes, I did	7 (5%)
No, they brought me to the vaccination center	133 (95%)
What was your last-received vaccination for? ^4^	
I cannot remember	23 (16.4%)
Meningococcus	21 (15%)
DTPa-IPV	2 (1.4%)
MPR	6 (4.2%)
HPV	9 (8.4%)
Pneumococcus	1 (0.01%)
Influenza	9 (8.4%)

^1^ 28 missing responses; ^2^ 8 missing responses; ^3^ 1 missing response; ^4^ 69 missing responses.

**Table 3 vaccines-11-01524-t003:** PMT score before and after the intervention.

Questionnaire Items	Students Who Increased Score after Intervention	*p*-Value
Perceived severity increase	47 (18.4%)	0.004
Perceived susceptibility increase	39 (15.2%)	0.03
Self-efficacy increase	38 (14.8%)	<0.001
Response efficacy increase	61 (23.8%)	0.03
Response cost increase	25 (9.8%)	<0.001
Maladaptive response rewards	35 (13.7%)	<0.001
Intention to vaccination increase	54 (21.1%)	0.002

**Table 4 vaccines-11-01524-t004:** Univariable and multivariable analyses of adolescents’ characteristics associated with a higher increase in PMT score.

Questionnaire Items	Crude OR	95% CI	*p*	Adjusted OR	95% CI	*p*
Age						
>13 years old	ref			ref		
<13 years old	1.3	0.7–2.3	0.34	1.5	0.3–6.7	0.60
Sex						
Female	ref			ref		
Male	0.7	0.4–1.3	0.26	0.9	0.5–1.7	0.84
School grade of lower secondary school						
1st grade	ref			ref		
2nd grade	1.2	0.5–3	0.63	0.4	0.1–1.8	0.22
3rd grade	1.2	0.7–2.2	0.49	0.3	0.1–1.6	0.16
Age of parents						
<44 years	ref			ref		
≥44 years	0.7	0.4–1.3	0.27	0.8	0.4–1.5	0.46
Location of the household						
Rural area	ref				ref	
Urban area	0.1	0.1–1	0.05	0.2	0.1–1.5	0.11
School dropout rates						
Very High	ref			ref		
High	3.1	1.3–7.5	0.008	8.3	1.7–40.8	0.009
Medium	2.3	1.1–5.1	0.04	2.7	1.1–6.7	0.03
Low	1.6	0.7–3.5	0.18	4.5	1.2–17.0	0.03
Can you remember the last vaccination you received?						
No	ref			ref		
Yes	0.94	0.5–1.7	0.84	1.5	0.9–2.5	0.09
Log likelihood = −126.7						

## Data Availability

The data will be made available after a motivated request to the corresponding author.

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
