# Peer review of "Changes in Students’ Perceptions Regarding Adolescent Vaccinations through a Before–After Study Conducted during the COVID-19 Pandemic: GIRASOLE Project Study"

_vaccines, 2023, doi:10.3390/vaccines11101524_

Round 1

Reviewer 1 Report

Authors focused in questionnaire-based behavioral study in order to study changes in student´s perceptions regarding adolescent vaccination, and in particular they addressed GIRASOLE project study. Several issues are missing, and before any positive recommendation can be given, the following major comments should be addressed by several revision. 

Sincerely, Referee 

Major comments.

Major comment 1: How much is protection motivation theory (PMT) suitable for the study? 

PMT does not explain whether the factors are part of threat or coping appraisals. In addition, the theory does not explain the relative importance of threat and coping appraisals when it comes to decision-making processes.

Major Comment 2:  Authors stated that " Overall, 256 out of 427 participating students answered the survey, with a response  

rate of 60.0%." If the response rate is 60%, 

what is the quality of analysis with respect to obtained relationship? It is too small sample size and too high non-response rate to speak about success of the intervention. Path analysis should be made and path diagram added to the study in order to better explain relevance of obtained relationships. 

Major Comment 3: There is no discussion of bias caused by application of A Multivariate Imputation by Chained Equations (MICE)  to deal with 189

the missing data in the sample. this should be added, accompanied with serious discussion amendments.  

Referee is missing goodness-of-fit for   the proportional odds model. 

Major comment 4: I am missing more detailed description of the multivariate logistic regression model, including goodness-of-fit tests in the given data setup. If authors used the only those statistically significant variables associated with an increase in the PMT score after the intervention in comparison to children who did not increase their score, how the induced dependence have been analysed? This condition is not building a standard  multivariate logistic regression model. 

Major comment 5: Conclusions are too terse.  Much more should be reported here.

it is ok

Author Response

Authors focused in questionnaire-based behavioral study in order to study changes in student´s perceptions regarding adolescent vaccination, and in particular they addressed GIRASOLE project study. Several issues are missing, and before any positive recommendation can be given, the following major comments should be addressed by several revision. 

Sincerely, Referee 

Major comments.

Major comment 1: How much is protection motivation theory (PMT) suitable for the study? 

PMT does not explain whether the factors are part of threat or coping appraisals. In addition, the theory does not explain the relative importance of threat and coping appraisals when it comes to decision-making processes.

 Answer: Thank you for your in-depth analysis and your useful comments. You have raised an important point. However, the Protection Motivation Theory (PMT) framework involves threat appraisal and coping appraisal as the multidimensional determinants of motivation. Surveys based on the constructs of Protection Motivation Theory have been extensively used in literature, also with COVID-19 prevention, to explain threat and coping appraisal when the health practices have the purpose to prevent infectious diseases, as you ca see in Lahiri A, et al. Role of Threat and Coping Appraisal in Protection Motivation for Adoption of Preventive Behavior During COVID-19 Pandemic. Front Public Health. 2021;9:678566. doi: 10.3389/fpubh.2021.678566. Protection Motivation Theory, indeed, emphasizes the factors that can lead to motivate and perform a preventive behavior.

Major Comment 2:  Authors stated that " Overall, 256 out of 427 participating students answered the survey, with a response rate of 60.0%." If the response rate is 60%, what is the quality of analysis with respect to obtained relationship? It is too small sample size and too high non-response rate to speak about success of the intervention. Path analysis should be made and path diagram added to the study in order to better explain relevance of obtained relationships. 

Answer: We appreciated the comment and used the occasion to revise what have already written in the manuscript. Indeed the 427 adolescent filled in at least one of the before or after questionnaire but not both questionnaire. However, 256 filled in both before and after questionnaire and were eligible for the analysis. In order to make more clear the inclusion process we added a new inclusion criteria in the method section “(iv) to fill in the questionnaire before and after the intervention”. Among the 256 questionnaires collected, there was several items with no-answer that forced us to opt for a multivariate imputation strategy.

Major Comment 3: There is no discussion of bias caused by application of A Multivariate Imputation by Chained Equations (MICE)  to deal with 189 the missing data in the sample. this should be added, accompanied with serious discussion amendments.  

Referee is missing goodness-of-fit for the proportional odds model. 

Answer: MICE operates under the assumption that given the variables used in the imputation procedure, the missing data are Missing At Random (MAR), which means that the probability that a value is missing depends only on observed values and not on unobserved values. In the MICE procedure a series of regression models are run whereby each variable with missing data is modelled conditional upon the other variables in the data. This means that each variable can be modelled according to its distribution, with, for example, binary variables modelled using logistic regression and continuous variables modelled using linear regression.

Comparison between missing (estimated by MICE) and no-missing data on other variables show no correlation between students with complete questionnaires and those with almost one missing answer in the questionnaire.

Available
PMT data
n. (%)

Missing data on PMT
(MICE imputation)
n. (%)

p-value of Fisher’s test

Sex

0.411

Male

37 (27.0%)

100 (73.0%)

Female

38 (31.9%)

81 (68.1%)

How old were you when you last performed a vaccination?

0.736

<5 years

2 (12.5%)

14 (87.5%)

5-10 years

14 (22.2%)

49 (77.8%)

>15 years

14 (23.3%)

46 (76.7%)

(missing)

45 (38.5%)

72 (61.5%)

Did you ask your parents to bring you to the vaccination centre?

>0.999

No, they brought me to the vaccination centre

29 (21.8%)

104 (78.2%)

Yes, I did

1 (14.3%)

6 (85.7%)

(missing)

45 (38.8%)

71 (61.2%)

Multiple imputation provide an appropriate measures of precision for these missing data incorporating relevant auxiliary variables. We added the point in the limitation section.

 Major comment 4: I am missing more detailed description of the multivariate logistic regression model, including goodness-of-fit tests in the given data setup. If authors used the only those statistically significant variables associated with an increase in the PMT score after the intervention in comparison to children who did not increase their score, how the induced dependence have been analysed? This condition is not building a standard  multivariate logistic regression model. 

Answer: We revised the text to describe the methodology adopted for the regression analysis: we have performed a stepwise selection (backward) to identify variables included in the final multivariable model adding the potential confounders. The response variable for the models was children with increased PMT score (delta between first and second questionnaire) versus the otherwise (delta less or equal to zero). So, it is no expected particular bias or induced dependence into the analyses following this approach. In the text we added the following sentence “The multivariable logistic regression model of factor associated to PMT increase before and after intervention used a backward stepwise selection to include other associated variables in according to the Likelihood ratio test for different models.”

Major comment 5: Conclusions are too terse.  Much more should be reported here.

Answer: We agree with you and have incorporated this suggestion, including new insights in the conclusions as following “School educational programs should focus on enhancing the teenagers’ knowledge and attitudes about vaccine-preventable diseases (VPDs), but at the same time they should address the vaccine-related fear and anxiety deriving from short- and long-term adverse effect of vaccinations. For this reason, stronger communication efforts and school education opportunities are required to promote vaccine decision-making among adolescents. Indeed, this study was performed in a period of mistrust and lack of confidence vaccines, mostly related to the contradictory information spread by media.

Furthermore, improving healthcare workers communication is one of the most highly prioritized goals when it comes to ensure timely and complete vaccination of youth, especially for those adolescents who have a vulnerable socio-economic status. Providers, especially in primary care settings, should be trained in effective vaccine communication as well as receive education regarding vaccine recommendations, contraindications and common parental and teenagers concerns. The evidence derived from this study highlights the importance of building tailored educational interventions to reduce disparities in vaccination intention among adolescents”.

Reviewer 2 Report

This is a very intersting before-after study with an educational intervention regarding knowledge and attitudes towards vaccinations in adolescents. The manuscript is well written, clear and concise. The Authors address the very relevant topic of social inequities in vaccination and Vaccine hesitancy.

I suggest just a couple of minor revisions on the first and second paragraphs of the conclusion, mainly because they doesn't seem deeply related to the present study.

The authors stated that - Line 338 - "The findings provided by this study suggest that vaccination knowledge improvement in an online setting among teenagers was strictly associated with their socio-economic status. Furthermore, adolescents need specific communication regarding the fear of needles and adverse reactions.".

The socio-economic status is not directly investigated in the study, but - as stated in the discussion - it is related to the early school leaving rates measured in the questionnaire as "School dropout index". The sentence in the asbtract seems more appropriate.

Furthermore the conclusion on the fear of needles and adverse reaction seems more a general conclusion than something really supported by the present study (Just 9% of the students increase their score after the intervention). On the contrary, the Authors could underline the quite high percentage of adolescents which could not remember the last vaccination performed (Approximately half of the surveyed students (56.5%, n=140))

Line 90:"first grade secondary schools" --> lower secondary schools

Line 204 - The authors sould revise table 2. They stated "Among the subgroup who remembered...". According to the previous question, the total observations should be 108, while in the table there are 16+63+60=139. Probably also the following quistion have the same problem (7+133=140)

On the contrary the total answers of the question "What was your last-performed vaccination for?" are 23+21+2+6+9+1+9=70 (including 23 "dont'remember").

Furthermore, I suggest to improve the bibliografy with a couple of Italian study on the topic:

Line 227- Add a reference. "There has been a wide body of research in the past years studying the efficacy of pre- and post- educational classes about vaccination in adolescents, focusing mainly on HPV, Tdap-IPV, MMR, and influenza [19]. The Study by Poscia A, et al (2019) "The impact of a school-based multicomponent intervention for promoting vaccine uptake in Italian adolescents: a retrospective cohort study", has been carried out in Italian secondary school.

Line 314 - Add a reference. "Similarly, the administration of vaccines in school settings should be considered since this method is associated with higher completion rates and raises opportunities to vaccinate adolescents, a group with lower GP or pediatrician attendance than other age groups." (i.e. the study of Desiante F, et al 2017. "Universal proposal strategies of anti-HPV vaccination for adolescents: comparative analysis between school-based and clinic immunization programs" demonstrated this association)

Author Response

This is a very intersting before-after study with an educational intervention regarding knowledge and attitudes towards vaccinations in adolescents. The manuscript is well written, clear and concise. The Authors address the very relevant topic of social inequities in vaccination and Vaccine hesitancy.

 I suggest just a couple of minor revisions on the first and second paragraphs of the conclusion, mainly because they doesn't seem deeply related to the present study.

The authors stated that - Line 338 - "The findings provided by this study suggest that vaccination knowledge improvement in an online setting among teenagers was strictly associated with their socio-economic status. Furthermore, adolescents need specific communication regarding the fear of needles and adverse reactions." The socio-economic status is not directly investigated in the study, but - as stated in the discussion - it is related to the early school leaving rates measured in the questionnaire as "School dropout index". The sentence in the asbtract seems more appropriate.

Answer: Thank you for your time and effort in reviewing our manuscript. The feedback has been invaluable in improving the content of the conclusion as following ”School educational programs should focus on enhancing the teenagers’ knowledge and attitudes about vaccine-preventable diseases (VPDs), but at the same time they should address the vaccine-related fear and anxiety deriving from short- and long-term adverse effect of vaccinations. For this reason, stronger communication efforts and school education opportunities are required to promote vaccine decision-making among adolescents. Indeed, this study was performed in a period of mistrust and lack of confidence vaccines, mostly related to the contradictory information spread by media.

Furthermore, improving healthcare workers communication is one of the most highly prioritized goals when it comes to ensure timely and complete vaccination of youth, especially for those adolescents who have a vulnerable socio-economic status. Providers, especially in primary care settings, should be trained in effective vaccine communication as well as receive education regarding vaccine recommendations, contraindications and common parental and teenagers concerns. The evidence derived from this study high-lights the importance of building tailored educational interventions to reduce disparities in vaccination intention among adolescents.”

Furthermore the conclusion on the fear of needles and adverse reaction seems more a general conclusion than something really supported by the present study (Just 9% of the students increase their score after the intervention). On the contrary, the Authors could underline the quite high percentage of adolescents which could not remember the last vaccination performed (Approximately half of the surveyed students (56.5%, n=140))

 Answer: We improved the conclusion section focusing on knowledge of vaccine and involvement in vaccination decision making of adolescent as following “The findings provided by this study highlights teenagers’ lack of knowledge about the VPDs, poor involvement in vaccination decision making and suggest that vaccination knowledge improvement in an online setting among teenagers was strictly associated with the school dropout rate of their neighborhood.” Additionally, in the conclusion we have replaced “socio-economic context” with “school dropout rate”.

Line 90:"first grade secondary schools" --> lower secondary schools

Answer: At the line 90 we have replaced “first grade” with “lower”.

Line 204 - The authors should revise table 2. They stated "Among the subgroup who remembered...". According to the previous question, the total observations should be 108, while in the table there are 16+63+60=139. Probably also the following quistion have the same problem (7+133=140)

On the contrary the total answers of the question "What was your last-performed vaccination for?" are 23+21+2+6+9+1+9=70 (including 23 "dont'remember").

Answer: As for the Table 2, we have added the missing values. The questions “How old were you when you last performed a vaccination?”. “Did you ask your parents to bring you to the vaccination center” and “What was your last-performed vaccination for” have been asked only to the 140 participants who answered “yes” to the question “Can you remember the last vaccination you performed?”, so the percentages are calculated on the 140, not on the total 256.

Furthermore, I suggest to improve the bibliografy with a couple of Italian study on the topic:

Line 227- Add a reference. "There has been a wide body of research in the past years studying the efficacy of pre- and post- educational classes about vaccination in adolescents, focusing mainly on HPV, Tdap-IPV, MMR, and influenza [19]. The Study by Poscia A, et al (2019) "The impact of a school-based multicomponent intervention for promoting vaccine uptake in Italian adolescents: a retrospective cohort study", has been carried out in Italian secondary school.

Line 314 - Add a reference. "Similarly, the administration of vaccines in school settings should be considered since this method is associated with higher completion rates and raises opportunities to vaccinate adolescents, a group with lower GP or pediatrician attendance than other age groups." (i.e. the study of Desiante F, et al 2017. "Universal proposal strategies of anti-HPV vaccination for adolescents: comparative analysis between school-based and clinic immunization programs" demonstrated this association)

Answer: We thank you for the suggestions about the references, which have been added as the reference number 19 and the reference number 40.

Reviewer 3 Report

The introduction provides a brief introduction to vaccine acceptance in certain regions in Italy and in some other countries including Australia where the HPV vaccine was developed. The authors outline the challenges with vaccine acceptance in vulnerable groups that include adolescents and they mention the vulnerable groups that are susceptible to misinformation. They effectively justify their reasons for targeting adolescents in this particular study.

The survey instruments appear to have been well designed with additional care being taken with the meaning of phrases so that while some questions were scored for answers to have decreasing scores, others involved increasing scores.

 In materials and methods the authors comprehensively describe the construction of the survey instruments, the target adolescent participants and the timing of delivery of the pre- and post- intervention questionnaire.

The PMT model used previously only in adults was employed in this study with adolescents participating as volunteers with an agreed undertaking to provide feedback, via questionnaires, throughout the study.

The authors provide a comprehensive outline of the PMT (Figure 1) and justify the choice of this model as the basis of the study.

Correct the spelling of Threat appraisal box in Path analysis summary Figure 1.

The chosen methods for statistical analysis appear to be appropriate for a study of this type and, clearly, the authors have sought thoughtful expert guidance. The updated changes appearing in the last sentence of Statistical analysis appear to be very specific, important and appropriate.

The path analysis (Fig 2) provides an interesting summary of the predictors of the protecting motivation.

It is important to check that the student numbers are adequate for an analysis of this type (Table 3) and I believe that numbers are adequate for reliable statistical analysis and justification of outcomes.

The conclusions drawn from this study are carefully and critically considered in the discussion. The findings are treated in an interesting way and effectively compared with findings from other studies that sought to determine factors of importance in vaccine hesitancy. Factors such as physical isolation, disadvantaged socio-economic, socio-cultural disadvantage, misinformation and restricted access to reliable and accurate information sources are critically discussed.

The discussion is good and it suggests where funding might be effectively committed to reduce fear and increase vaccine coverage so this adds significantly to our understanding of vaccine hesitancy in adolescents in Palermo. This reviewer can recall a paper (Vaccines - circa late 2021 if I recall correctly)  that compared vaccine hesitancy in Palermo and a rural isolated area of Sicily with that in Bologna and, if this paper has not already been cited, it might add to the value of this paper in addressing this very important issue of vaccine hesitancy.

This version appears to have been nicely updated in response to comments from previous reviewers and the sections appearing in red greatly improve the clarity of the sentences.

Many of the students could not recall their specific vaccination records (nor can this reviewer!) and I did wonder whether vaccination records in Italy are recorded centrally as is the case in some countries. If these records are centrally recorded maybe, with permission of students and parents/guardians, these vaccination records might be made available for ethically approved research projects of this type.

Well done, a nice informative paper that adds to our understanding of factors influencing vaccine hesitancy in adolescents.

Minor points

L27 Maybe delete in order which adds nothing to the sentence

L51 Being teenagers among those affected the most, -- Maybe change to  Since teenagers were among the most affected, there is a need to reach higher vaccination coverage ….

L76 The primary objective of this study was to evaluate,

L171 Suggest deleting …in order…  Commonly used in sentences but what do these two words add to the sentence except length?

Author Response

We are very glad to the new reviewer because from their comments it seems that he/she read in a very complete manner the manuscript.

We checked the grammar spelling in the Figure 1 and the other minor spelling suggested.

We also added the reference suggested by reviewer in discussion section.

Round 2

Reviewer 1 Report

Unfortunatelly, Authors did not addressed satisfactory my major comments.  the following major comments should be addressed by several revision.

Sincerely, Referee

Major comments.

Major comment 1: How much is protection motivation theory (PMT) suitable for the study? Authors needs to reply to this question in a more comprehensive way. 

  The theory does not explain the relative importance of threat and coping appraisals when it comes to decision-making processes. References to published studies and careful discussion on limitations is needed. 

Major Comment 2:  Again, I need to state that   with a response

rate of 60.0%. 

authors are reaching a too low sample size.  It is too small sample size and too high non-response rate to speak about success of the intervention. Path analysis should be made and path diagram added to the study in order to better explain relevance of obtained relationships. 

Referee does not agree that "adopting a Multivariate Imputation by Chained Equations provide an appropriate 331

measures of precision for these missing data incorporating relevant auxiliary variables", proper analysis and justification of this textual statement is completely missing. As Referee requested in Major Comment 3: There is no discussion of bias caused by application of A Multivariate Imputation by Chained Equations (MICE)  to deal with 189

the missing data in the sample. this should be added, accompanied with serious discussion amendments.

Referee is missing goodness-of-fit for   the proportional odds model.

Major comment 4: I am missing more detailed description of the multivariate logistic regression model, including goodness-of-fit tests in the given data setup. If authors used the only those statistically significant variables associated with an increase in the PMT score after the intervention in comparison to children who did not increase their score, how the induced dependence have been analysed? This condition is not building a standard  multivariate logistic regression model.

it is withing a limits

Author Response

Unfortunatelly, Authors did not addressed satisfactory my major comments.  the following major comments should be addressed by several revision.

Sincerely, Referee

Major comments.

Major comment 1: How much is protection motivation theory (PMT) suitable for the study? Authors needs to reply to this question in a more comprehensive way. The theory does not explain the relative importance of threat and coping appraisals when it comes to decision-making processes. References to published studies and careful discussion on limitations is needed. 

Answer: The PMT theory is one of the most commonly used theories to explain how individuals take health-promoting measures and in the last years it was used in a large quantity of studies in order to evaluate determinants of vaccination intentions as COVID-19, influenza, HPV etc.

According to your suggestions, we discussed more in depth the role of threat and coping appraisal of PMT in the discussion section as following: “Besides, international literature suggests that coping appraisal typically has greater influence on motivation intention than threat appraisal [12]. This research provides further evidence of the importance of self-efficacy, perceived response cost (p<0.001) and response efficacy (p=0.003) in predicting intention to vaccinate, consistently with previous research that has applied these coping appraisal constructs to the prediction of vaccination uptake [16]. Self-efficacy is considered a great predictor of health-related behavior change and maintenance [20]. Similarly, in this study self-efficacy is one of the main drivers of vaccine acceptability and a lower role for severity of disease perception. This can be explained by a lower importance in according to adolescent way of thinking of possible consequence of VPD as showed also for COVID vaccination among adolescent [21]. Furthermore, in the present survey, response efficacy is associated with vaccine availability, affordability, and accessibility. However, the main barrier in Italy is not economic, since teenagers’ vaccinations are offered actively free of charge, but may be related to the discontinued provision of immunization services during COVID-19 pandemic [1]. However, the key barriers to receiving teenagers’ vaccination appear to be concerns about side effects and safety.”

Major Comment 2:  Again, I need to state that  with a response rate of 60.0%. authors are reaching a too low sample size.  It is too small sample size and too high non-response rate to speak about success of the intervention. Path analysis should be made and path diagram added to the study in order to better explain relevance of obtained relationships. 

Answer: Thank you very much for this helpful suggestion. According to your comments we performed a Path analysis in order to evaluate the relevance of obtained relationships. The resulting path diagram was showed in the result section as following “As showed in the figure 2, the path analysis of the PMT score showed that the factor with a higher coefficient was coping appraisal b=0.750 on PMT score increase, followed by coping appraisal b=0.598 on intention and threat appraisal b=0.536 on PMT score increase.”

With respect to the sample size we are aware that a response rate of 60% could be a main limitation of our study and we reported this consideration in the Discussion section. In this sense we can not exclude that the not responding 40% could adopt different models for explaining their decision-making processes. However, it should be also considered that our results could explain the decision-making process of at least 60% of the study population and thus, at our opinion, they should be considered still of great interest to the international scientific community.

Referee does not agree that "adopting a Multivariate Imputation by Chained Equations provide an appropriate measures of precision for these missing data incorporating relevant auxiliary variables", proper analysis and justification of this textual statement is completely missing. As Referee requested in Major Comment 3: There is no discussion of bias caused by application of A Multivariate Imputation by Chained Equations (MICE)  to deal with 189 the missing data in the sample. this should be added, accompanied with serious discussion amendments.

Referee is missing goodness-of-fit for   the proportional odds model.

Answer: We agree with you that multivariate imputation could suffer from some important limitation including the risk of biasing analyses. However it is also true that this is a very commonly used statistical method that can help in improving the quality of analyses. According to your comments, in the manuscript we clarified and improved the explanation about the choice to use this methods and reported that the Multiple Imputation by Chained Equation is a method with few solutions to evaluate the validity of analyses [44]. One of the main approach is to use a cross validation method. In detail, we firstly evaluated the predictive value for items of PMT using the imputed database. This analysis showed a low predictive value for the following variables: “perceived susceptibility” among before intervention questionnaires and “intention” among the after intervention questionnaires (see Figure below).

Furthermore, we performed for these variables the Lipsitz goodness of fit test for ordinal response models.  According to this test, the goodness of fit for “perceived susceptibility” variable among before questionnaires had a Likelihood Ratio= 14.45, df = 9, p-value = 0.107. Moreover, for “intention” among the after questionnaires was showed a Likelihood Ratio = 5.25, df = 9, p-value = 0.812. Finally, we can assume that the MICE analysis provided appropriate measures of precision for these missing data incorporating relevant auxiliary variables.

We hope the these sentences could be helpful for improving the manuscript and we are available for considering further modifications if required from you or from the Editor.

Major comment 4: I am missing more detailed description of the multivariate logistic regression model, including goodness-of-fit tests in the given data setup. If authors used the only those statistically significant variables associated with an increase in the PMT score after the intervention in comparison to children who did not increase their score, how the induced dependence have been analysed? This condition is not building a standard  multivariate logistic regression model.

Answer: According to the referee request we revised the multivariable model selection and selected the best fitting model which had a Log likelihood= -126.7, a Likelihood ratio chi-square test=19.11 with probability>chi-square=0.04. This model had a similar value for the OR that you can find in the new table 4 in the Adjusted OR in comparison to the old Table 4. Furthermore, the associated factor is always the school dropouts rates whichever is the model, adding robustness to the meaning of the multivariable analysis. 

Round 3

Reviewer 1 Report

Unfortunatelly, Authors still did not addressed satisfactory my major comments.  the following serious  comments should be addressed by several revision.

Sincerely, Referee

Major comments.

Major comment 1: How much is protection motivation theory (PMT) suitable for the study? Authors needs to reply to this question in a more comprehensive way.  Authors did not reply how   the  PMT in their context     explains the relative importance of threat and coping appraisals when it comes to decision-making processes. Comparisons to Covid-19 and HPV are not carefully provided. This should be done. References to published studies and careful discussion on limitations is needed.

Major Comment 2:  I am sorry but  I again need to state that   with a response rate of 60.0% authors are reaching a too low sample size. This disqualifies the main part of study.  It is too small sample size and too high non-response rate to speak about success of the intervention. Path analysis was added and shows a  small difference in betas for copying appraisal and for treat appraisal, namely 0.75, and 0.536. You need to also publish both standard deviations   for both beta-coefficients, jointly with their difference statistics, in order to say there is a difference. 

 Major Comment 3: Unfortunately, authors still failed to justify usage of MICE, and this introduces bias in the study.  Quality of analysis cannot be improved by  usage of MICE without further regularity checks.  This should be added, accompanied with serious discussion amendments. Referee is missing goodness-of-fit for   the proportional odds model, and Lipsitz, LR based data you put to answers are not convincing. Since you have many missings, you need to apply Robust regression model for ordinal response, e.g. the one developed in DOI: https://dx.doi.org/10.4310/20-SII631.

Major comment 4: Again, unfortunately the authors failed to reply to my comment. You cannot use standard Likelihood and chi^2 tests, since your data are principally dependent. I am missing a more detailed description of the multivariate logistic regression model, including goodness-of-fit tests in the given data setup. If authors used only those statistically significant variables associated with an increase in the PMT score after the intervention in comparison to children who did not increase their score, how the induced dependence have been analysed? This condition is not building a standard  multivariate logistic regression model.  Moreover  the associated factor is always the school dropouts rates whichever model is not  adding robustness to the meaning of the multivariable analysis, unless it is statistically proven. 

OK

Author Response

Unfortunatelly, Authors still did not addressed satisfactory my major comments.  the following serious  comments should be addressed by several revision.

Sincerely, Referee

Major comments.

Major comment 1: How much is protection motivation theory (PMT) suitable for the study? Authors needs to reply to this question in a more comprehensive way.  Authors did not reply how   the  PMT in their context     explains the relative importance of threat and coping appraisals when it comes to decision-making processes. Comparisons to Covid-19 and HPV are not carefully provided. This should be done. References to published studies and careful discussion on limitations is needed.

Response: We explained the suitability of PMT for this study in the discussion section. Indeed the results of path analysis indicated that PMT is a good framework to predict the adoption of preventive behaviour, especially because PMT constructs are significantly correlated with intention to receive a vaccine and to the increase of PMT score. As a consequence, the present model is suitable for the present study. The effective constructs on predicting vaccination intention were response efficacy, self-efficacy (directly), and perceived costs (reversely). It is thus recommended to employ this theory and its constructs to design interventional programs to promote vaccinations.

Moreover we underlined the relationship among treat and coping appraisal in decision making process in the discussion section. The decision-making process is influenced by intention, which is one of the PMT variables that we investigated in the survey and is the most proximal predictor of behaviour, according to the Protection Motivation Theory. In the present study, greater intentions were significantly associated with higher coping appraisal (Coef. = 0.598, p<00.01) and threat appraisal (Coef. = 0.312, p=0.03). Threat appraisal and coping appraisal are the multidimensional determinants of intention, according to PMT framework, and affect individuals’ motivation to take preventive behaviour.

Major Comment 2:  I am sorry but  I again need to state that   with a response rate of 60.0% authors are reaching a too low sample size. This disqualifies the main part of study.  It is too small sample size and too high non-response rate to speak about success of the intervention. Path analysis was added and shows a  small difference in betas for copying appraisal and for treat appraisal, namely 0.75, and 0.536. You need to also publish both standard deviations   for both beta-coefficients, jointly with their difference statistics, in order to say there is a difference. 

 Response: The size of our final sample can be considered suitable since the population of the municipality of Palermo in the age groups 10-17 is 46,136. Thus, the minimum sample size needed to have a 95% confidence level that the true value is within ±5% is 138 or 245, assuming 10% or 20% of efficacy of intervention, respectively at the population level.We have specified that estimation are obtained by robust method and added standard error for each beta-coefficients estimation. 

Major Comment 3: Unfortunately, authors still failed to justify usage of MICE, and this introduces bias in the study.  Quality of analysis cannot be improved by  usage of MICE without further regularity checks.  This should be added, accompanied with serious discussion amendments. Referee is missing goodness-of-fit for   the proportional odds model, and Lipsitz, LR based data you put to answers are not convincing. Since you have many missings, you need to apply Robust regression model for ordinal response, e.g. the one developed in DOI: https://dx.doi.org/10.4310/20-SII631.

Response: We apologize in worst previous explanation, we had only few missing data in our dataset for most of variables, specifically:-) 1 or 2 missing response in “response_cost”, “response_efficacy” and “self_efficacy”;-) 5 missing response in “maladaptive”-) 80 in “intention”-) 90 in “susceptibility”The total number of items response possible was 256 multiplied by 6 items for each questionnaire by 2 response (before and after), with a total of 3,072 responses. We believe that the starting missing records have not a high level. However, we have already clarify the good results in testing of intention and susceptibility. We used the cross validation method and the Lipsitz goodness of fit test as reported in another manuscript. Furthermore, robust variant of weighted least squares using the Satorra-Bentler scaled test statistic method to correct the model and to produce robust standard errors.

Major comment 4: Again, unfortunately the authors failed to reply to my comment. You cannot use standard Likelihood and chi^2 tests, since your data are principally dependent. I am missing a more detailed description of the multivariate logistic regression model, including goodness-of-fit tests in the given data setup. If authors used only those statistically significant variables associated with an increase in the PMT score after the intervention in comparison to children who did not increase their score, how the induced dependence have been analysed? This condition is not building a standard  multivariate logistic regression model.  Moreover  the associated factor is always the school dropouts rates whichever model is not  adding robustness to the meaning of the multivariable analysis, unless it is statistically proven. 

Response: We understand that our data should be interpreted as dependent because they were obtained from a before after study. Consequently, we tried to use a technique for correlated data such as the random effect logistic regression analysis. However, the rho value obtained from that model was 0.82. This higher value make us doubtful about reliability of our estimate. On the other hand we explored another solution, consisting in calculating a single outcome measure for each pair of before after questionnaire and we based the analysis on this summary measure. We added a clarification of the outcome used in the analysis in the methods section “… the PMT score before and after the intervention, that was a summary measure of the before after questionnaires for each subject.” This is an option allowed to avoid to believe that statistical evidence can be stronger than it really is (S. Rabe-Hesketh Multilevel and longitudinal modeling using Stata).